# General Graph Random Features

**Isaac Reid**[1*]**, Krzysztof Choromanski**[2,3*]**, Eli Berger**[4*]**, Adrian Weller**[1,5]
[1]University of Cambridge, [2]Google DeepMind, [3]Columbia University,
[4]University of Haifa, [5]Alan Turing Institute
`ir337@cam.ac.uk, kchoro@google.com`

## Abstract

We propose a novel random walk-based algorithm for unbiased estimation of arbitrary functions of a weighted adjacency matrix, coined *general graph random features* (g-GRFs). This includes many of the most popular examples of kernels defined on the nodes of a graph. Our algorithm enjoys subquadratic time complexity with respect to the number of nodes, overcoming the notoriously prohibitive cubic scaling of exact graph kernel evaluation. It can also be trivially distributed across machines, permitting learning on much larger networks. At the heart of the algorithm is a *modulation function* which upweights or downweights the contribution from different random walks depending on their lengths. We show that by parameterising it with a neural network we can obtain g-GRFs that give higher-quality kernel estimates or perform efficient, scalable kernel learning. We provide robust theoretical analysis and support our findings with experiments including pointwise estimation of fixed graph kernels, solving non-homogeneous graph ordinary differential equations, node clustering and kernel regression on triangular meshes.[1]

## 1 Introduction and related work

The *kernel trick* is a powerful technique to perform nonlinear inference using linear learning algorithms (Campbell, 2002; Kontorovich et al., 2008; Canu and Smola, 2006; Smola and Schölkopf, 2002). Supposing we have a set of $N$ datapoints $\mathcal{X} = \{\boldsymbol{x}_i\}_{i=1}^N$, it replaces Euclidean dot products $\boldsymbol{x}_i^\top \boldsymbol{x}_j$ with evaluations of a *kernel function* $K : \mathcal{X} \times \mathcal{X} \to \mathbb{R}$, capturing the 'similarity' of the datapoints by instead taking an inner product between implicit (possibly infinite-dimensional) feature vectors in some Hilbert space $\mathbb{H}_K$.

An object of key importance is the *Gram matrix* $\mathbf{K} \in \mathbb{R}^{N \times N}$ whose entries enumerate the pairwise kernel evaluations, $\mathbf{K} := [K(\boldsymbol{x}_i, \boldsymbol{x}_j)]_{i,j=1}^N$. Despite the theoretical rigour and empirical success enjoyed by kernel-based learning algorithms, the requirement to manifest and invert this matrix leads to notoriously poor $\mathcal{O}(N^3)$ time-complexity scaling. This has spurred research dedicated to efficiently approximating $\mathbf{K}$, the chief example of which is *random features* (Rahimi and Recht, 2007): a Monte-Carlo approach which gives explicitly manifested, finite dimensional vectors $\phi(\boldsymbol{x}_i) \in \mathbb{R}^m$ whose Euclidean dot product is equal to the kernel evaluation in expectation,

$$\mathbf{K}_{ij} = \mathbb{E}\left[\phi(\boldsymbol{x}_i)^\top \phi(\boldsymbol{x}_j)\right]. \tag{1}$$

This allows one to construct a *low-rank decomposition* of $\mathbf{K}$ which provides much better scalability. Testament to its utility, a rich taxonomy of random features exists to approximate many different Euclidean kernels including the Gaussian, softmax, and angular and linear kernels (Johnson, 1984; Dasgupta et al., 2010; Goemans and Williamson, 2004; Choromanski et al., 2020).

Kernels defined on discrete input spaces, e.g. $K : \mathcal{N} \times \mathcal{N} \to \mathbb{R}$ with $\mathcal{N}$ the set of nodes of a graph $\mathcal{G}$ (Smola and Kondor, 2003; Kondor and Lafferty, 2002; Chung and Yau, 1999),

---

[*]Equal contribution.
[1]Code is available at https://github.com/isaac-reid/general_graph_random_features.

enjoy widespread applications including in bioinformatics (Borgwardt et al., 2005), community detection (Kloster and Gleich, 2014) and recommender systems (Yajima, 2006). More recently, they have been used in applications as diverse as manifold learning for deep generative modelling (Zhou et al., 2020) and for solving single- and multiple-source shortest path problems (Crane et al., 2017). However, for these graph-based kernel methods the problem of poor scalability is particularly acute. This is because even computing the corresponding Gram matrix $\mathbf{K}$ is typically of at least cubic time complexity in the number of nodes $N$, requiring e.g. the inversion of an $N \times N$ matrix or computation of multiple matrix-matrix products. Despite the presence of this computational bottleneck, random feature methods for graph kernels have proved elusive. Indeed, only recently was a viable *graph random feature* (GRF) mechanism proposed by Choromanski (2023). Their algorithm uses an ensemble of random walkers which deposit a 'load' at every vertex they pass through that depends on i) the product of weights of edges traversed by the walker and ii) the marginal probability of the subwalk. Using this scheme, it is possible to construct random features $\{\phi(i)\}_{i=1}^{N} \subset \mathbb{R}^{N}$ such that $\phi(i)^{\top}\phi(j)$ gives an unbiased approximation to the $ij$-th matrix element of the 2-regularised Laplacian kernel. Multiple independent approximations can be combined to estimate the $d$-regularised Laplacian kernel with $d \neq 2$ or the diffusion kernel (although the latter is only asymptotically unbiased). The GRFs algorithm enjoys both subquadratic time complexity and strong empirical performance on tasks like $k$-means node clustering, and it is trivial to distribute across machines when working with massive graphs.

However, a key limitation of GRFs is that they only address a limited family of graph kernels which may not be suitable for the task at hand. Our central contribution is a simple modification which generalises the algorithm to *arbitrary functions of a weighted adjacency matrix*, allowing efficient and unbiased approximation a much broader class of graph node kernels. We achieve this by introducing an extra *modulation function $f$* that controls each walker's load as it traverses the graph. As well as empowering practitioners to approximate many more fixed kernels, we demonstrate that $f$ can also be parameterised by a neural network and learned. We use this powerful approach to optimise g-GRFs for higher-quality approximation of fixed kernels and for scalable implicit kernel learning.

The remainder of the manuscript is organised as follows. In **Sec. 2** we introduce *general graph random features* (g-GRFs) and prove that they enable scalable approximation of *arbitrary* functions of a weighted adjacency matrix, including many of the most popular examples of kernels defined on the nodes of a graph. We also extend the core algorithm with *neural modulation functions*, replacing one component of the g-GRF mechanism with a neural network, and derive generalisation bounds for the corresponding class of learnable graph kernels (**Sec. 2.1**). In **Sec. 3** we run extensive experiments, including: pointwise estimation of a variety of popular graph kernels (**Sec. 3.1**); simulation of time evolution under non-homogeneous graph ordinary differential equations (**Sec. 3.2**); kernelised $k$-means node clustering including on large graphs (**Sec. 3.3**); training a neural modulation function to suppress the mean square error of fixed kernel estimates (**Sec 3.4**); and training a neural modulation function to learn a kernel for node attribute prediction on triangular mesh graphs (**Sec. 3.5**).

## 2 General graph random features

Consider a directed weighted graph $\mathcal{G}(\mathcal{N}, \mathcal{E}, \mathbf{W} \coloneqq [w_{ij}]_{i,j \in \mathcal{N}})$ where: i) $\mathcal{N} \coloneqq \{1, ..., N\}$ is the set of nodes; ii) $\mathcal{E}$ is the set of edges, with $(i, j) \in \mathcal{E}$ if there is a directed edge from $i$ to $j$ in $\mathcal{G}$; and iii) $\mathbf{W}$ is the *weighted adjacency matrix*, with $w_{ij}$ the weight of the directed edge from $i$ to $j$ (equal to 0 if no such edge exists). Note that an undirected graph can be described as directed with the symmetric weighted adjacency matrix $\mathbf{W}$. Now consider the matrices $\mathbf{K}_{\boldsymbol{\alpha}}(\mathbf{W}) \in \mathbb{R}^{N \times N}$, where $\boldsymbol{\alpha} = (\alpha_k)_{k=0}^{\infty}$ and $\alpha_k \in \mathbb{R}$:

$$\mathbf{K}_{\boldsymbol{\alpha}}(\mathbf{W}) = \sum_{k=0}^{\infty} \alpha_k \mathbf{W}^k. \tag{2}$$

We assume that the sum above converges for all $\mathbf{W}$ under consideration, which can be ensured with a regulariser $\mathbf{W} \to \sigma \mathbf{W}$, $\sigma \in \mathbb{R}_+$. Without loss of generality, we also assume

that $\boldsymbol{\alpha}$ is *normalised* such that $\alpha_0 = 1$. The matrix $\mathbf{K}_{\boldsymbol{\alpha}}(\mathbf{W})$ can be associated with a graph function $K_{\boldsymbol{\alpha}}^{\mathcal{G}} : \mathcal{N} \times \mathcal{N} \to \mathbb{R}$ mapping from a pair of graph nodes to a real number.

Note that if $\mathcal{G}$ is an undirected graph then $\mathbf{K}_{\boldsymbol{\alpha}}(\mathbf{W})$ automatically inherits the symmetry of $\mathbf{W}$. In this case, it follows from Weyl's perturbation inequality (Bai et al., 2000) that $\mathbf{K}_{\boldsymbol{\alpha}}(\mathbf{W})$ is positive semidefinite for any given $\boldsymbol{\alpha}$ provided the spectral radius $\rho(\mathbf{W}) :=$ $\max_{\lambda \in \Lambda(\mathbf{W})} (|\lambda|)$ is sufficiently small (with $\Lambda(\mathbf{W})$ the set of eigenvalues of $\mathbf{W}$). This can again be ensured by multiplying the weight matrix $\mathbf{W}$ by a regulariser $\sigma \in \mathbb{R}_+$. It then follows that $\mathbf{K}_{\boldsymbol{\alpha}}(\mathbf{W})$ can be considered the Gram matrix of a *graph kernel function $K_{\boldsymbol{\alpha}}^{\mathcal{G}}$*.

With suitably chosen $\boldsymbol{\alpha} = (\alpha_k)_{k=0}^{\infty}$, the class described by Eq. 2 includes many popular examples of graph node kernels in the literature (Smola and Kondor, 2003; Chapelle et al., 2002). They measure connectivity between nodes and are typically functions of the *graph Laplacian matrix*, defined by $\mathbf{L} := \mathbf{I} - \widetilde{\mathbf{W}}$ with $\widetilde{\mathbf{W}} := [w_{ij}/\sqrt{d_i d_j}]_{i,j=1}^N$. Here, $d_i := \sum_j w_{ij}$ is the weighted degree of node $i$ such that $\widetilde{\mathbf{W}}$ is the *normalised* weighted adjacency matrix. For reference, Table 1 gives the kernel definitions and normalised coefficients $\alpha_k$ (corresponding to powers of $\widetilde{\mathbf{W}}$) to be considered later in the manuscript. In practice, factors in $\alpha_k$ equal to a quantity raised to the power of $k$ are absorbed into the normalisation of $\widetilde{\mathbf{W}}$.

| Name | Form | $\alpha_k$ |
|------|------|------------|
| $d$-regularised Laplacian | $(\mathbf{I}_N + \sigma^2 \mathbf{L})^{-d}$ | $\binom{d+k-1}{k}\left(1+\sigma^{-2}\right)^{-k}$ |
| $p$-step random walk | $(\alpha \mathbf{I}_N - \mathbf{L})^p, \alpha \geq 2$ | $\binom{p}{k}(\alpha-1)^{-k}$ |
| Diffusion | $\exp(-\sigma^2 \mathbf{L}/2)$ | $\frac{1}{k!}(\frac{\sigma^2}{2})^k$ |
| Inverse Cosine | $\cos(\mathbf{L}\pi/4)$ | $\frac{1}{k!}\left(\frac{\pi}{4}\right)^k \cdot \left\{(-1)^{\frac{k}{2}} \text{ if } k \text{ even}, (-1)^{\frac{k-1}{2}} \text{ if } k \text{ odd}\right\}$ |

Table 1: Different graph functions/kernels $K_{\boldsymbol{\alpha}}^{\mathcal{G}} : \mathcal{N} \times \mathcal{N} \to \mathbb{R}$. The exp and cos mappings are defined via Taylor series expansions rather than element-wise, e.g. $\exp(\mathbf{M}) := \lim_{n \to \infty}(\mathbf{I}_N + \mathbf{M}/n)^n$ and $\cos(\mathbf{M}) := \mathrm{Re}(\exp(i\mathbf{M}))$. $\sigma$ and $\alpha$ are regularisers. Note that the diffusion kernel is sometimes instead defined by $\exp(\sigma^2(\mathbf{I}_N - \mathbf{L}))$ but these forms are equivalent up to normalisation. Also note that the $p$-step random walk kernel is closely related to the graph Matérn kernel (Borovitskiy et al., 2021).

The chief goal of this work is to construct a *random feature map* $\phi(i) : \mathcal{N} \to \mathbb{R}^l$ with $l \in \mathbb{N}$ that provides unbiased approximation of $\mathbf{K}_{\boldsymbol{\alpha}}(\mathbf{W})$ as in Eq. 1. To do so, we consider the following algorithm.

---

**Algorithm 1** Constructing a random feature vector $\phi_f(i) \in \mathbb{R}^N$ to approximate $\mathbf{K}_{\boldsymbol{\alpha}}(\mathbf{W})$

---

**Input:** weighted adjacency matrix $\mathbf{W} \in \mathbb{R}^{N \times N}$, vector of unweighted node degrees (no. neighbours) $\boldsymbol{d} \in \mathbb{R}^N$, modulation function $f : (\mathbb{N} \cup \{0\}) \to \mathbb{R}$, termination probability $p_{\text{halt}} \in (0,1)$, node $i \in \mathcal{N}$, number of random walks to sample $m \in \mathbb{N}$.
**Output:** random feature vector $\phi_f(i) \in \mathbb{R}^N$

1: initialise: $\phi_f(i) \leftarrow \mathbf{0}$
2: **for** $w = 1, ..., m$ **do**
3:     initialise: `load` $\leftarrow 1$
4:     initialise: `current_node` $\leftarrow i$
5:     initialise: `terminated` $\leftarrow$ False
6:     initialise: `walk_length` $\leftarrow 0$
7:     **while** `terminated` $=$ False **do**
8:         $\phi_f(i)[\texttt{current\_node}] \leftarrow \phi_f(i)[\texttt{current\_node}] + \texttt{load} \times f(\texttt{walk\_length})$
9:         `walk_length` $\leftarrow$ `walk_length` $+1$
10:         `new_node` $\leftarrow \mathrm{Unif}[\mathcal{N}(\texttt{current\_node})]$          ▷ assign to one of neighbours
11:         `load` $\leftarrow$ `load` $\times \frac{d[\texttt{current\_node}]}{1-p_{\text{halt}}} \times \mathbf{W}[\texttt{current\_node}, \texttt{new\_node}]$          ▷ update load
12:         `current_node` $\leftarrow$ `new_node`
13:         `terminated` $\leftarrow (t \sim \mathrm{Unif}(0,1) < p_{\text{halt}})$      ▷ draw RV $t$ to decide on termination
14:     **end while**
15: **end for**
16: normalise: $\phi_f(i) \leftarrow \phi_f(i)/m$

---

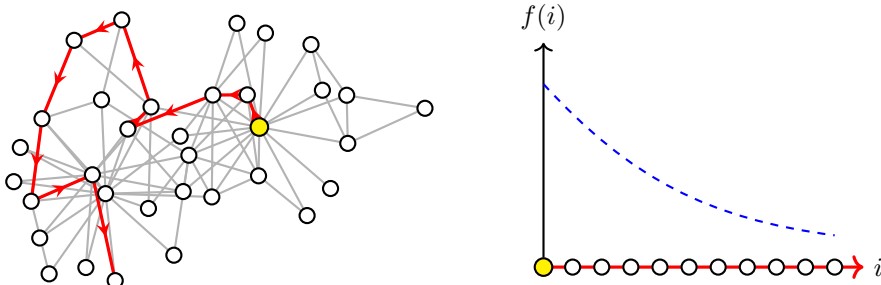

Figure 1: Schematic for a random walk on a graph (solid red) and an accompanying modulation function $f$ (dashed blue) used to approximate an arbitrary graph node function $K^{\mathcal{G}}$.

This is identical to the algorithm presented by Choromanski (2023) for constructing features to approximate the 2-regularised Laplacian kernel, apart from the presence of the extra *modulation function* $f : (\mathbb{N} \cup \{0\}) \to \mathbb{R}$ in line 8 that upweights or downweights contributions from walks depending on their length (see Fig. 1). We refer to $\phi_f$ as *general graph random features* (g-GRFs), where the subscript $f$ identifies the modulation function. Crucially, *the time complexity of Alg. 1 is subquadratic in the number of nodes $N$*, in contrast to exact methods which are $\mathcal{O}(N^3)$.[2]

We now state the following central result, proved in App. A.2.

**Theorem 2.1** (Unbiased approximation of $K_{\boldsymbol{\alpha}}$ via convolutions). *For two modulation functions: $f_1, f_2 : (\mathbb{N} \cup \{0\}) \to \mathbb{R}$, g-GRFs $\left(\phi_{f_1}(i)\right)_{i=1}^N, \left(\phi_{f_2}(i)\right)_{i=1}^N$ constructed according to Alg. 1 give unbiased approximation of $\mathbf{K}_{\boldsymbol{\alpha}}$,*

$$[\mathbf{K}_{\boldsymbol{\alpha}}]_{ij} = \mathbb{E}\left[\phi_{f_1}(i)^\top \phi_{f_2}(j)\right], \tag{3}$$

*for kernels with an arbitrary Taylor expansion $\boldsymbol{\alpha} = (\alpha_k)_{k=0}^\infty$ provided that $\boldsymbol{\alpha} = f_1 * f_2$. Here, $*$ is the discrete convolution of the modulation functions $f_1, f_2$; that is, for all $k \in (\mathbb{N} \cup \{0\})$,*

$$\sum_{p=0}^k f_1(k-p) f_2(p) = \alpha_k. \tag{4}$$

Clearly the class of pairs of modulation functions $f_1, f_2$ that satisfy Eq. 4 is highly degenerate. Indeed, it is possible to solve for $f_1$ given any $f_2$ and $\boldsymbol{\alpha}$ provided $f_2(0) \neq 0$. For instance, a trivial solution is given by: $f_1(i) = \alpha_i, \quad f_2(i) = \mathbb{I}(i = 0)$ with $\mathbb{I}(\cdot)$ the indicator function. In this case, the walkers corresponding to $f_2$ are 'lazy', depositing all their load at the node at which they begin. Contributions to the estimator $\phi_{f_1}(i)^\top \phi_{f_2}(j)$ only come from walkers initialised at $i$ traversing all the way to $j$ rather than two walkers both passing through an intermediate node. Also of great interest is the case of *symmetric modulation functions $f_1 = f_2$*, where now intersections do contribute. In this case, the following is true (proved in App. A.3).

**Theorem 2.2** (Computing symmetric modulation functions). *Supposing $f_1 = f_2 = f$, Eq. 4 is solved by a function $f$ which is unique (up to a sign) and is given by*

$$f(i) = \pm \sum_{n=0}^i \binom{\frac{1}{2}}{n} \sum_{\substack{k_1 + 2k_2 + 3k_3 \dots = i \\ k_1 + k_2 + k_3 + \dots = n}} \binom{n}{k_1 k_2 k_3 \dots} \left(\alpha_1^{k_1} \alpha_2^{k_2} \alpha_3^{k_3} \dots\right). \tag{5}$$

*Moreover, $f(i)$ can be efficiently computed with the iterative formula*

$$\begin{cases} f(0) = \pm\sqrt{\alpha_0} = \pm 1 \\ f(i+1) = \frac{\alpha_{i+1} - \sum_{p=0}^{i-1} f(i-p) f(p+1)}{2f(0)} & \text{for } i \geq 0. \end{cases} \tag{6}$$

[2]Concretely, Alg. 1 yields a pair of matrices $\boldsymbol{\phi}_{1,2} := (\phi(i))_{i=1}^N \in \mathbb{R}^{N \times N}$ such that $\mathbf{K} = \mathbb{E}(\boldsymbol{\phi}_1 \boldsymbol{\phi}_2^\top)$ in subquadratic time. Of course, explicitly multiplying the matrices to evaluate every element of $\widehat{\mathbf{K}}$ would be $\mathcal{O}(N^3)$, but we avoid this since in applications we just evaluate $\boldsymbol{\phi}_1(\boldsymbol{\phi}_2^\top \boldsymbol{v})$ where $\boldsymbol{v} \in \mathbb{R}^N$ is some vector and the brackets give the order of computation. This is $\mathcal{O}(N^2)$.

For symmetric modulation functions, the random features $\phi_{f_1}(i)$ and $\phi_{f_2}(i)$ are identical apart from the particular sample of random walks used to construct them. They cannot share the same sample or estimates of diagonal kernel elements $[\mathbf{K}_{\boldsymbol{\alpha}}]_{ii}$ will be biased.

**Computational cost**: Note that when running Alg. 1 one only needs to evaluate the modulation functions $f_{1,2}(i)$ up to the length of the longest walk one samples. A batch of size $b$, $(f_{1,2}(i))_{i=1}^b$, can be pre-computed in time $\mathcal{O}(b^2)$ and reused for random features corresponding to different nodes and even different graphs. Further values of $f$ can be computed at runtime if $b$ is too small and also reused for later computations. Moreover, the minimum length $b$ required to ensure that all $m$ walks are shorter than $b$ with high probability $(\Pr(\cup_{i=1}^m \text{len}(\omega_i) \leq b) > 1 - \delta, \ \delta \ll 1)$ scales only logarithmically with $m$ (see App. A.1). This means that, despite estimating a much more general family of graph functions, g-GRFs are essentially no more expensive than the original GRF algorithm. Moreover, any techniques used for dimensionality reduction of regular GRFs (e.g. applying the Johnson-Lindenstrauss transform (Dasgupta et al., 2010) or using 'anchor points' (Choromanski, 2023)) can also be used with g-GRFs, providing further efficiency gains.

**Generating functions:** Inserting the constraint for unbiasedness in Eq. 4 back into the definition of $\mathbf{K}_{\boldsymbol{\alpha}}(\mathbf{W})$, we immediately have that

$$\mathbf{K}_{\boldsymbol{\alpha}}(\mathbf{W}) = \mathbf{K}_{f_1}(\mathbf{W})\mathbf{K}_{f_2}(\mathbf{W}) \tag{7}$$

where $\mathbf{K}_{f_1}(\mathbf{W}) \coloneqq \sum_{i=0}^{\infty} f_1(i)\mathbf{W}^i$ is the *generating function* corresponding to the sequence $(f_1(i))_{i=0}^{\infty}$. This is natural because the (discrete) Fourier transform of a (discrete) convolution returns the product of the (discrete) Fourier transforms of the respective functions. In the symmetric case $f_1 = f_2$, it follows that

$$\mathbf{K}_f(\mathbf{W}) = \pm (\mathbf{K}_{\boldsymbol{\alpha}}(\mathbf{W}))^{\frac{1}{2}}. \tag{8}$$

If the RHS has a simple Taylor expansion (e.g. $\mathbf{K}_{\boldsymbol{\alpha}}(\mathbf{W}) = \exp(\mathbf{W})$ so $\mathbf{K}_f(\mathbf{W}) = \exp(\frac{\mathbf{W}}{2})$), this enables us obtain $f$ without recourse to the conditional sum in Eq. 5 or the iterative expression in Eq. 6. This is the case for many popular graph kernels; we provide some prominent examples in the table left. A notable exception is the inverse cosine kernel.

| Name | $f(i)$ |
|---|---|
| $d$-regularised Laplacian | $\frac{(d-2+2i)!!}{(2i)!!(d-2)!!}$ |
| $p$-step random walk | $\binom{\frac{p}{2}}{i}$ |
| Diffusion | $\frac{1}{2^i i!}$ |

As an interesting corollary, by considering the diffusion kernel we have also proved that $\sum_{p=0}^k \frac{1}{2^p p!} \frac{1}{2^{k-p}(k-p)!} = \frac{1}{k!}$.

## 2.1 NEURAL MODULATION FUNCTIONS, KERNEL LEARNING AND GENERALISATION

Instead of using a fixed modulation function $f$ to estimate a fixed kernel, it is possible to parameterise it more flexibly. For example, we can define a *neural modulation function* $f^{(N)} : (\mathbb{N} \cup \{0\}) \to \mathbb{R}$ by a neural network (with a restricted domain) whose input and output dimensionalities are equal to 1. During training, we can choose the loss function to target particular desiderata of g-GRFs: for example, to suppress the mean square error of estimates of some particular fixed kernel (Sec. 3.4), or to learn a kernel which performs better in a downstream task (Sec. 3.5). Implicitly learning $\mathbf{K}_{\boldsymbol{\alpha}}$ via $f^{(N)}$ is more scalable than learning $\mathbf{K}_{\boldsymbol{\alpha}}$ directly because it obviates the need to repeatedly compute the exact kernel, which is typically of $\mathcal{O}(N^3)$ time complexity. Since any modulation function $f$ maps to a unique $\boldsymbol{\alpha}$ by Eq. 4, it is also always straightforward to recover the exact kernel which the g-GRFs estimate, e.g. once the training is complete.

Supposing we have (implicitly) learned $\boldsymbol{\alpha}$, how can the learned kernel $K_{\boldsymbol{\alpha}}^{\mathcal{G}}$ be expected to generalise? Let $\boldsymbol{\psi}_{K_{\alpha}} : x \to \mathbb{H}_{K_{\alpha}}$ denote the feature mapping from the input space to the reproducing kernel Hilbert space $\mathbb{H}_{K_{\alpha}}$ induced by the kernel $K_{\boldsymbol{\alpha}}^{\mathcal{G}}$. Define the hypothesis set

$$H = \{x \to \boldsymbol{w}^{\top}\boldsymbol{\psi}_{K_{\alpha}}(x) : |\alpha_i| \leq \alpha_i^{(M)}, \|\boldsymbol{w}\|_2 \leq 1\}, \tag{9}$$

where we restricted our family of kernels so that the absolute value of each Taylor coefficient $\alpha_i$ is smaller than some maximum value $\alpha_i^{(M)}$. Following very similar arguments to Cortes et al. (2010), the following is true.

**Theorem 2.3** (Empirical Rademacher complexity bound). *For a fixed sample $S = (x_i)_{i=1}^m$, the empirical Rademacher complexity $\widehat{\mathcal{R}}(H)$ is bounded by*

$$\widehat{\mathcal{R}}(H) \leq \sqrt{\frac{1}{m} \sum_{i=0}^{\infty} \alpha_i^{(M)} \rho(\mathbf{W})^i}, \tag{10}$$

*where $\rho(\mathbf{W})$ is the spectral radius of the weighted adjacency matrix $\mathbf{W}$.*

Naturally, the bound on $\widehat{\mathcal{R}}(H)$ increases monotonically with $\rho(\mathbf{W})$. Following standard arguments in the literature, this immediately yields generalisation bounds for learning kernels $K_{\boldsymbol{\alpha}}^{\mathcal{G}}$. We discuss this in detail, including proving Theorem 2.3, in App. A.4.

## 3 Experiments

Here we test the empirical performance of g-GRFs, both for approximating fixed kernels (Secs 3.1-3.3) and with learnable neural modulation functions (Secs 3.4-3.5).

### 3.1 Unbiased pointwise estimation of fixed kernels

We begin by confirming that g-GRFs do indeed give unbiased estimates of the graph kernels listed in Table 1, taking regularisers $\sigma = 0.25$ and $\alpha = 20$ with $p_{\text{halt}} = 0.1$. We use symmetric modulation functions $f$, computed with the closed forms where available and using the iterative scheme in Eq. 6 if not. Fig. 2 plots the relative Frobenius norm error between the true kernels $\mathbf{K}$ and their approximations with g-GRFs $\widehat{\mathbf{K}}$ (that is, $\|\mathbf{K} - \widehat{\mathbf{K}}\|_F / \|\mathbf{K}\|_F$) against the number of random walkers $m$. We consider 8 different graphs: a small random Erdős-Rényi graph, a larger Erdős-Rényi graph, a binary tree, a $d$-regular graph and 4 real world examples (`karate`, `dolphins`, `football` and `eurosis`) (Ivashkin, 2023). They vary substantially in size. For every graph and for all kernels, the quality of the estimate improves as $m$ grows and becomes very small with even a modest number of walkers.

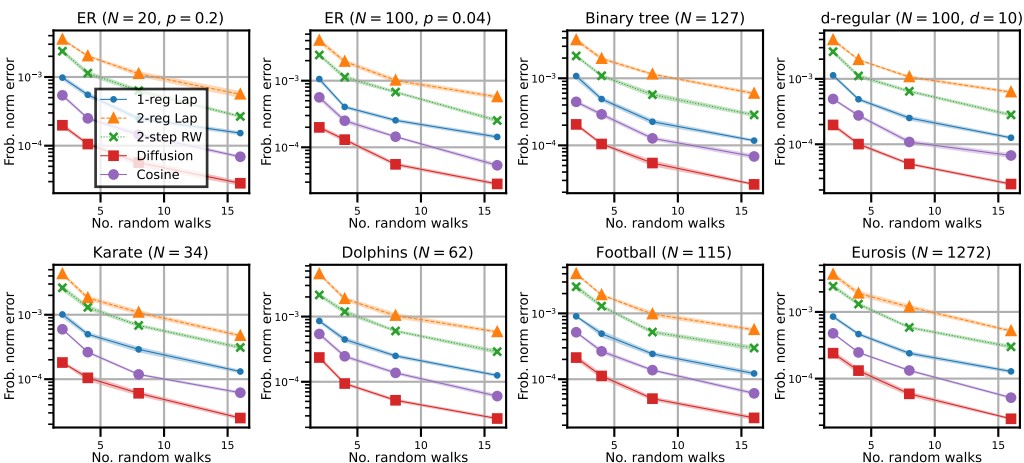

Figure 2: Unbiased estimation of popular kernels on different graphs using g-GRFs. The approximation error ($y$-axis) improves with the number of walkers ($x$-axis). We repeat 10 times; one standard deviation of the mean error is shaded.

### 3.2 Solving differential equations on graphs

An intriguing application of g-GRFs for fixed kernels is efficiently computing approximate solutions of time-invariant non-homogeneous ordinary differential equations (ODEs) on graphs. Consider the following ODE defined on the nodes $\mathcal{N}$ of the graph $\mathcal{G}$:

$$\frac{d\boldsymbol{x}(t)}{dt} = \mathbf{W}\boldsymbol{x}(t) + \boldsymbol{y}(t), \tag{11}$$

where $\boldsymbol{x}(t) \in \mathbb{R}^N$ is the state of the graph at time $t$, $\mathbf{W} \in \mathbb{R}^{N \times N}$ is a weighted adjacency matrix and $\boldsymbol{y}(t)$ is a (known) driving term. Assuming the null initial condition $\boldsymbol{x}(0) = \mathbf{0}$, Eq. 11 is solved by the convolution

$$\boldsymbol{x}(t) = \int_0^t \exp(\mathbf{W}(t-\tau))\boldsymbol{y}(\tau)d\tau = \mathbb{E}_{\tau \in \mathcal{P}}\left[\frac{1}{p(\tau)}\exp(\mathbf{W}(t-\tau))\boldsymbol{y}(\tau)\right] \quad (12)$$

where $\mathcal{P}$ is a probability distribution on the interval $[0,t]$, equipped with a (preferably efficient) sampling mechanism and probability density function $p(\tau)$. Taking $n \in \mathbb{N}$ Monte Carlo samples $(\tau_i)_{i=1}^n \overset{\text{i.i.d.}}{\sim} \mathcal{P}$, we can construct the unbiased estimator:

$$\widehat{\boldsymbol{x}}(t) := \frac{1}{n}\sum_{j=1}^n \frac{1}{p(\tau_j)}\exp(\mathbf{W}(t-\tau_j))\boldsymbol{y}(\tau_j). \quad (13)$$

Note that $\exp(\mathbf{W}(t-\tau_j))$ is nothing other than the diffusion kernel, which is expensive to compute exactly for large $N$ but can be efficiently approximated with g-GRFs. Take $\exp(\mathbf{W}(t-\tau_j)) \simeq \boldsymbol{\Phi}_j \boldsymbol{\Phi}_j^\top$ with $\boldsymbol{\Phi}_j := (\phi(i))_{i=1}^N$ an $N \times N$ matrix whose rows are g-GRFs constructed to approximate the kernel at a particular $\tau_j$. Then we have that

$$\widehat{\boldsymbol{x}}(t) := \frac{1}{n}\sum_{j=1}^n \frac{1}{p(\tau_j)}\boldsymbol{\Phi}_j \boldsymbol{\Phi}_j^\top \boldsymbol{y}(\tau_j) \quad (14)$$

which can be computed in quadratic time (c.f. cubic if the heat kernel is computed exactly). Further speed gains are possible if dimensionality reduction techniques are applied to the g-GRFs (Choromanski, 2023; Dasgupta et al., 2010).

As an example, we consider diffusion on three real-world graphs with a fixed source at one node, taking $\mathbf{W} = \mathbf{L}$ (the graph Laplacian) and $\boldsymbol{y}(t) = \boldsymbol{y} = (1, 0, 0, ...)^\top$. The steady state is $\boldsymbol{x}(\infty) = \mathbf{W}^{-1}(-\boldsymbol{y})$. We simulate evolution under the ODE for $t = 1$ with $n = 10$ discretisation timesteps and $\mathcal{P}$ uniform, approximating $\exp(\mathbf{W}(t-\tau_j))$ with different numbers of walkers $m$. As $m$ grows, the quality of approximation improves and the (normalised) error on the final state $\|\widehat{\boldsymbol{x}}(1) - \boldsymbol{x}(1)\|_2/\|\boldsymbol{x}(1)\|_2$ drops for every graph. We take 100 repeats for statistics and plot the results in Fig. 3. One standard deviation of the mean error is shaded. $p_{\text{halt}} = 0.1$ and the regulariser is given by $\sigma = 1$.

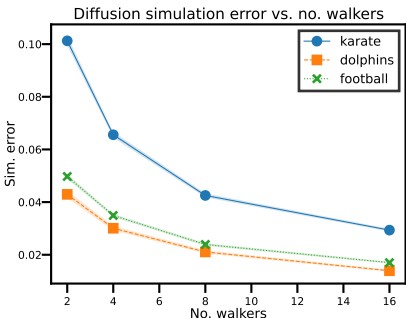

Figure 3: ODE simulation error decreases as the number of walkers grows.

### 3.3 Efficient kernelised graph node clustering

As a further demonstration of the utility of our new mechanism, we show how estimates of the kernel $\mathbf{K} = \exp(\sigma^2\mathbf{W})$ can be used to assign nodes to $k = 3$ clusters. Here, we choose $\mathbf{W}$ to be the (unweighted) adjacency matrix and the regulariser is $\sigma^2 = 0.2$. We follow the algorithm proposed by Dhillon et al. (2004), comparing the clusters when we use exact and g-GRF-approximated kernels. For all graphs, $m \leq 80$. Table 2 reports the clustering error, defined by

$$E_c := \frac{\text{no. wrong pairs}}{N(N-1)/2}. \quad (15)$$

Table 2: Errors in kernelised $k$-means clustering when approximating the kernel $\exp(\sigma^2\mathbf{W})$ with g-GRFs.

| Graph | $N$ | Clustering error, $E_c$ |
|---|---|---|
| karate | 34 | 0.08 |
| dolphins | 62 | 0.16 |
| polbooks | 105 | 0.12 |
| football | 115 | 0.02 |
| databases | 1046 | 0.10 |
| eurosis | 1272 | 0.09 |
| cora | 2485 | 0.01 |
| citeseer | 3300 | 0.04 |

This is the number of misclassified pairs of nodes (assigned to the same cluster when the converse is true or vice versa) divided by the total number of pairs. The error is small even with a modest number of walkers and on large graphs; kernel estimates efficiently constructed using g-GRFs can be readily deployed on downstream tasks where exact methods are slow.

### 3.4 Learning $f^{(N)}$ for better kernel approximation

Following the discussion in Sec. 2.1 we now replaced fixed $f$ with a *neural modulation function* $f^{(N)}$ parameterised by a simple neural network with 1 hidden layer of size 1:

$$f^{(N)}(x) = \sigma_{\text{softplus}}\left(w_2\sigma_{\text{ReLU}}(w_1x + b_1) + b_2\right), \tag{16}$$

where $w_1, b_1, w_2, b_2 \in \mathbb{R}$ and $\sigma_{\text{ReLU}}$ and $\sigma_{\text{softplus}}$ are the ReLU and softplus activation functions, respectively. Bigger, more expressive architectures (including allowing $f^{(N)}$ to take negative values) can be used but this is found to be sufficient for our purposes.

We define our loss function to be the Frobenius norm error between a target Gram matrix and our g-GRF-approximated Gram matrix on the small Erdős-Rényi graph ($N = 20$) with $m = 16$ walks. For the target, we choose the 2-regularised Laplacian kernel. We train symmetric $f_1^{(N)} = f_2^{(N)}$ but provide a brief discussion of the asymmetric case (including its respective strengths and weaknesses) in App. A.5. On this graph, we minimise the loss with the Adam optimiser and a decaying learning rate (LR $= 0.01, \gamma = 0.975$, 1000 epochs). We make the following striking observation: $f^{(N)}$ does *not* generically converge to the unique unbiased (symmetric) modulation function implied by $\boldsymbol{\alpha}$, but instead to some different $f$ that though biased gives a smaller mean squared error (MSE). This is possible because by downweighting long walks the learned $f^{(N)}$ gives estimators with a smaller variance, which is sufficient to suppress the MSE on the kernel matrix elements even though it no longer gives the target value in expectation. We then fix $f^{(N)}$ and use it for kernel approximation on the remaining graphs. The learned, biased $f^{(N)}$ still provides better kernel estimates, including for graphs with very different topologies and a much greater number of nodes: `eurosis` is bigger by a factor of over 60. See Table 3 for the results. $p_{\text{halt}} = 0.5$ and $\sigma = 0.8$.

Naturally, the learned $f^{(N)}$ is dependent on the the number of random walks $m$; as $m$ grows, the variance on the kernel approximation drops so it is intuitive that the learned $f^{(N)}$ will approach the unbiased $f$. Fig. 4 empirically confirms this is the case, showing the learned $f^{(N)}$ for different numbers of walkers. The line labelled $\infty$ is the unbiased modulation function, which for the 2-regularised Laplacian kernel is constant.

Table 3: Kernel approximation error with $m = 16$ walks and unbiased or learned modulation functions. Lower is better. Brackets give one standard deviation of the last digit.

Figure 4: Learned modulation function with different numbers of random walkers $m$. It approaches the unbiased $f^{(N)}$ as $m \to \infty$

| Graph | $N$ | Frob. norm error on $\widehat{K}$ | |
|---|---|---|---|
| | | Unbiased | Learned |
| Small ER | 20 | 0.0488(9) | **0.0437(9)** |
| Larger ER | 100 | 0.0503(4) | **0.0448(4)** |
| Binary tree | 127 | 0.0453(4) | **0.0410(4)** |
| $d$-regular | 100 | 0.0490(2) | **0.0434(2)** |
| karate | 34 | 0.0492(6) | **0.0439(6)** |
| dolphins | 62 | 0.0505(5) | **0.0449(5)** |
| football | 115 | 0.0520(2) | **0.0459(2)** |
| eurosis | 1272 | 0.0551(2) | **0.0484(2)** |

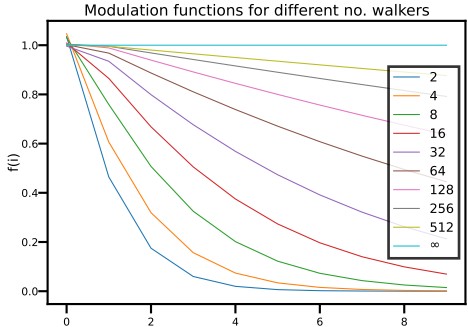

These learned modulation functions might guide the more principled construction of biased, low-MSE GRFs in the future. An analogue in Euclidean space is provided by structured orthogonal random features (SORFs), which replace a random Gaussian matrix with a **HD**-product to estimate the Gaussian kernel (Choromanski et al., 2017; Yu et al., 2016). This likewise improves kernel approximation quality despite breaking estimator unbiasedness.

### 3.5 Implicit kernel learning for node attribute prediction

As suggested in Sec. 2.1, it is also possible to train the neural modulation function $f^{(N)}$ directly using performance on a downstream task, performing *implicit kernel learning*. We have argued that this is much more scalable than optimising $\mathbf{K}_{\boldsymbol{\alpha}}$ directly.

In this spirit, we now address the problem of kernel regression on triangular mesh graphs, as previously considered by Reid et al. (2023). For graphs in this dataset (Dawson-Haggerty, 2023), every node is associated with a normal vector $\boldsymbol{v}^{(i)} \in \mathbb{R}^3$ equal to the mean of the normal vectors of its 3 surrounding faces. The task is to predict the directions of missing vectors (a random 5% split) from the remainder. Our (unnormalised) predictions are given by $\widehat{\boldsymbol{v}}^{(i)} := \sum_j \widehat{\mathbf{K}}^{(N)}(i,j)\boldsymbol{v}^{(j)}$, where $\widehat{\mathbf{K}}^{(N)}$ is a kernel estimate constructed using g-GRFs with a neural modulation function $f^{(N)}$ (see Eq. 3). The angular prediction error is $1 - \cos\theta_i$ with $\theta_i$ the angle between the true $\boldsymbol{v}^{(i)}$ and and

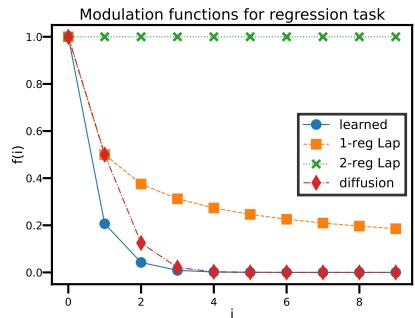

approximate $\widehat{\boldsymbol{v}}^{(i)}$ normals, averaged over the missing vectors. We train a symmetric pair $f^{(N)}$ using this angular prediction error on the small `cylinder` graph ($N = 210$) as the loss function. Then we freeze $f^{(N)}$ and compute the angular prediction error for other larger graphs. Fig. 5 shows the learned $f^{(N)}$ as well as some other modulation functions corresponding to popular fixed kernels. Note also that learning $f^{(N)}$ already includes (but is not limited to) optimising the lengthscale of a given kernel: taking $\widetilde{\mathbf{W}} \to \beta\widetilde{\mathbf{W}}$ is identical to $f(i) \to f(i)\beta^i \ \forall \ i$.

Figure 5: Fixed and learned modulation functions for kernel regression

The prediction errors are highly correlated between the different modulation functions for a given random draw of walks; ensembles which 'explore' the graph poorly, terminating quickly or repeating edges, will give worse g-GRF estimators for every $f$. For this reason, we compute the prediction errors as the average normalised *difference* compared to the learned kernel result. Table 4 reports the results. Crucially, *this difference is found to be positive for every graph and every fixed kernel*, meaning the learned kernel always performs best.

Table 4: Normalised difference in angular prediction error compared to the learned kernel defined by $f^{(N)}$ (trained on the `cylinder` graph). All entries are positive since the learned kernel always performs best. We take 100 repeats (but only 10 for the very large `cycloidal` graph). The brackets give one standard deviation on the final digit.

| Graph | $N$ | Normalised $\Delta$(pred error) c.f. learned | | |
| --- | --- | --- | --- | --- |
| | | 1-reg Lap | 2-reg Lap | Diffusion |
| cylinder | 210 | +0.40(2) | +0.85(4) | +0.029(3) |
| teapot | 480 | +0.81(5) | +1.78(8) | +0.059(3) |
| idler-riser | 782 | +0.52(2) | +1.12(1) | +0.042(2) |
| busted | 1941 | +0.81(2) | +1.60(4) | +0.063(2) |
| torus | 4350 | +2.13(5) | +5.3(1) | +0.067(2) |
| cycloidal | 21384 | +0.065(4) | +0.13(1) | +0.011(1) |

It is remarkable that the learned $f^{(N)}$ gives the smallest error for all the graphs even though it was just trained on `cylinder`, the smallest one. We have implicitly learned a good kernel for this task which generalises well across topologies. It is also intriguing that the diffusion kernel performs only slightly worse. This is to be expected because their modulation functions are similar (see Fig. 5) so they encode very similar kernels, but this will not always be the case depending on the task at hand.

## 4 Conclusion

We have introduced 'general graph random features' (g-GRFs), a novel random walk-based algorithm for time-efficient estimation of arbitrary functions of a weighted adjacency matrix. The mechanism is conceptually simple and trivially distributed across machines, unlocking kernel-based machine learning on very large graphs. By parameterising one component of the random features with a simple neural network, we can futher suppress the mean square error of estimators and perform scalable implicit kernel learning.

## 5 Ethics and reproducibility

**Ethics**: Our work is foundational. There are no direct ethical concerns that we can see, though of course increases in scalability afforded by graph random features might amplify risks of graph-based machine learning, from bad actors or as unintended consequences.

**Reproducibility**: To foster reproducibility, we clearly state the central algorithm in Alg. 1. Source code is available at https://github.com/isaac-reid/general_graph_random_features. All theoretical results are accompanied by proofs in Appendices A.1-A.4, where any assumptions are made clear. The datasets we use correspond to standard graphs and are freely available online. We link suitable repositories in every instance. Except where prohibitively computationally expensive, results are reported with uncertainties to help comparison.

## 6 Relative contributions and acknowledgements

EB initially proposed using a modulation function to generalise GRFs to estimate the diffusion kernel and derived its mathematical expression. IR and KC then developed the full g-GRFs algorithm for general functions of a weighted adjacency matrix, proving all theoretical results, running the experiments and preparing the manuscript. AW provided helpful discussions and advice.

IR acknowledges support from a Trinity College External Studentship. AW acknowledges support from a Turing AI fellowship under grant EP/V025279/1 and the Leverhulme Trust via CFI.

We thank Kenza Tazi and Austin Tripp for their careful readings of the manuscript. Richard Turner provided valuable suggestions and support throughout the project.

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

## A    APPENDICES

### A.1    MINIMUM BATCH SIZE SCALES LOGARITHMICALLY WITH NUMBER OF WALKS

Consider an ensemble of $m$ random walks $S \coloneqq \{\omega_i\}_{i=1}^{m}$ whose lengths are sampled from a geometric distribution with termination probability $p$. Trivially, $\Pr(\text{len}(\omega_i) < b) = 1 - (1 - p)^b$. Given $m$ independent walkers,

$$\Pr\left(\max_{\omega_i \in S}(\text{len}(\omega_i) < b\right) = \Pr\left(\cup_{i=1}^{m} \text{len}(\omega_i) < b\right) = (1 - (1-p)^b)^m. \tag{17}$$

Take 'with high probability' to mean with probability at least $1 - \delta$, with $\delta \ll 1$ fixed. Then

$$b = \frac{\log(1 - (1 - \delta)^{\frac{1}{m}})}{\log(1-p)} \simeq \frac{\log(\delta) - \log(m)}{\log(1-p)}. \tag{18}$$

As the number of walkers $m$ grows, the minimum value of $b$ to ensure that all walkers are shorter than $b$ with high probability scales logarithmically with $m$.

### A.2    PROOF OF THEOREM 2.1 (UNBIASED APPROXIMATION OF $K_\alpha$ VIA CONVOLUTIONS)

We begin by proving that g-GRFs constructed according to Alg. 1 give unbiased estimation of graph functions with Taylor coefficients $(\alpha_i)_{i=0}^{\infty}$ provided the discrete convolution relation in Eq. 4 is fulfilled.

Denote the set of $m$ walks sampled out of node $i$ by $\{\bar{\Omega}_k^{(i)}\}_{k=1}^{m}$. Carefully considering Alg. 1, it is straightforward to convince oneself that the g-GRF estimator takes the form

$$\phi(i)_v \coloneqq \frac{1}{m} \sum_{k=1}^{m} \sum_{\omega_{iv} \in \Omega_{iv}} \frac{\widetilde{\omega}(\omega_{iv}) f(\text{len}(\omega_{iv}))}{p(\omega_{iv})} \mathbb{I}(\omega_{iv} \in \bar{\Omega}_k^{(i)}). \tag{19}$$

Here, $\Omega_{iv}$ denotes the set of all graph walks between nodes $i$ and $v$, of which each $\omega_{iv}$ is a member. $\text{len}(\omega_{iv})$ is a function that gives the length of walk $\omega_{iv}$ and $\widetilde{\omega}(\omega_{iv})$ evaluates the products of edge weights it traverses. $p(\omega_{iv}) = (1-p)^{\text{len}(\omega_{iv})} \prod_{i=1}^{\text{len}(\omega_{iv})} \frac{1}{d_i}$ denotes the walk's marginal probability. $\mathbb{I}$ is the indicator function which evaluates to 1 if its argument is true (namely, walk $\omega_{iv}$ is a prefix subwalk of $\bar{\Omega}_k^{(i)}$, the $k$th walk sampled out of $i$) and 0 otherwise. Trivially, we have that $\mathbb{E}\left[\mathbb{I}(\omega_{iv} \in \bar{\Omega}_k^{(i)})\right] = p(\omega_{iv})$ (by construction to make the estimator unbiased).[3]

Now note that:

$$\mathbb{E}\left[\phi(i)^\top \phi(j)\right] = \mathbb{E}\left[\sum_{v \in \mathcal{N}} \phi(i)_v \phi(j)_v\right]$$

$$= \sum_v \sum_{\omega_{iv} \in \Omega_{iv}} \sum_{\omega_{jv} \in \Omega_{jv}} \widetilde{\omega}(\omega_{iv}) f(\text{len}(\omega_{iv})) \widetilde{\omega}(\omega_{jv}) f(\text{len}(\omega_{jv}))$$

$$= \sum_v \sum_{l_1=0}^{\infty} \sum_{l_2=0}^{\infty} \mathbf{W}_{iv}^{l_1} \mathbf{W}_{jv}^{l_2} f(l_1) f(l_2) \tag{20}$$

$$= \sum_v \sum_{l_1=0}^{\infty} \sum_{l_3=0}^{l_1} \mathbf{W}_{iv}^{l_1-l_3} \mathbf{W}_{jv}^{l_3} f(l_1 - l_3) f(l_3)$$

$$= \sum_{l_1=0}^{\infty} \mathbf{W}_{ij}^{l_1} \sum_{l_3=0}^{l_1} f(l_1 - l_3) f(l_3).$$

---

[3]To flag a subtle point: in the sum we take $\omega_{iv} \in \Omega_{iv}$ to mean that $\omega_{iv}$ is a member of the *set of all possible walks* of any length between nodes $i$ and $v$, $\Omega_{iv}$. On the other hand, inside the indicator function by $\omega_{iv} \in \bar{\Omega}_k^{(i)}$ we mean that $\omega_{iv}$ is a prefix subwalk of *one particular walk* $\bar{\Omega}_k^{(i)}$ sampled out of node $i$. In these two cases the interpretation of the symbol $\in$ should be slightly different.

From the first to the second line we used the definition of GRFs in Eq. 19. We then rewrote the sum of the product of edge weights over all possible paths as powers of the weighted adjacency matrix $\mathbf{W}$, with $l_{1,2}$ corresponding to walk lengths. To get to the fourth line, we changed indices in the infinite sums, then the final line followed simply.

The final expression in Eq. 20 is exactly equal to $\mathbf{K_\alpha}(\mathbf{W}) \coloneqq \sum_{l_1=0}^{\infty} \alpha_{l_1} \mathbf{W}^{l_1}$ provided we have that

$$\sum_{l_3=0}^{l_1} f(l_1 - l_3)f(l_3) \coloneqq \alpha_{l_1}, \tag{21}$$

as stated in Eq. 4 of the main text (with variables renamed to $k$ and $p$).

### A.3 Proof of Theorem 2.2 (Computing symmetric modulation functions)

Here, we show how to compute $f$ under the constraint that the modulation functions are identical, $f_1 = f_2 = f$. We will use the relationship in Eq. 4, reproduced below for the reader's convenience:

$$\sum_{p=0}^{i} f(i-p)f(p) = \alpha_i. \tag{22}$$

The iterative form in Eq. 6 is easy to show. For $i = 0$ we have that $f(0)^2 = \alpha_0$, so $f(0) = \pm\sqrt{\alpha_0} = \pm 1$ (where we used the normalisation condition $\alpha_0 = 1$). Now note that

$$\sum_{p=0}^{i+1} f(i+1-p)f(p) = 2f(0)f(i+1) + \sum_{p=1}^{i} f(i+1-p)f(p) = \alpha_{i+1} \tag{23}$$

so clearly

$$f(i+1) = \frac{\alpha_{i+1} - \sum_{p=0}^{i-1} f(i-p)f(p+1)}{2f(0)}. \tag{24}$$

This enables us to compute $f(i+1)$ given $\alpha_{i+1}$ and $(f(p))_{p=0}^{i}$.

The analytic form in Eq. 5 is only a little harder. Inserting the discrete convolution relation in Eq. 4 back into Eq. 2, we have that

$$\mathbf{K_\alpha}(\mathbf{W}) = \mathbf{K}_{f_1}(\mathbf{W})\mathbf{K}_{f_2}(\mathbf{W}) \tag{25}$$

where $\mathbf{K}_{f_1}(\mathbf{W}) \coloneqq \sum_{i=0}^{\infty} f_1(i)\mathbf{W}^i$ is the *generating function* corresponding to the sequence $(f_1(i))_{i=0}^{\infty}$. Constraining $f_1 = f_2$,

$$\mathbf{K}_f(\mathbf{W}) = \pm (\mathbf{K_\alpha}(\mathbf{W}))^{\frac{1}{2}}. \tag{26}$$

We also discuss this in the 'generating functions' section of Sec. 2 where we use it to derive simple closed forms for $f$ for some special kernels. Now we have that

$$\begin{aligned}
\sum_{i=0}^{\infty} f(i)\mathbf{W}^i &= \pm \left(\sum_{n=0}^{\infty} \alpha_n \mathbf{W}^n\right)^{\frac{1}{2}} \\
&= \pm \left(1 + \left(\alpha_1 \mathbf{W} + \alpha_2 \mathbf{W}^2 + ...\right)\right)^{\frac{1}{2}} \\
&= \pm \sum_{n=0}^{\infty} \binom{\frac{1}{2}}{n} \left(\alpha_1 \mathbf{W} + \alpha_2 \mathbf{W}^2 + ...\right)^n.
\end{aligned} \tag{27}$$

We need to equate powers of $\mathbf{W}$ between the generating functions. Consider the terms proportional to $\mathbf{W}^i$. Clearly no such terms will feature when $n > i$, so we can restrict the sum on the RHS to $0 \leq n \leq i$. Meanwhile, the term in $\left(\alpha_1 \mathbf{W} + \alpha_2 \mathbf{W}^2 + ...\right)^n$ proportional to $\mathbf{W}^i$ is nothing other than

$$\sum_{\substack{k_1+2k_2+3k_3...=i \\ k_1+k_2+k_3+...=n}} \binom{n}{k_1 k_2 k_3 ...} \left(\alpha_1^{k_1} \alpha_2^{k_2} \alpha_3^{k_3} ...\right) \tag{28}$$

where $\binom{n}{k_1 k_2 k_3 \ldots}$ is the multinomial coefficient. Combining,

$$f(i) = \pm \sum_{n=0}^{i} \binom{\frac{1}{2}}{n} \sum_{\substack{k_1+2k_2+3k_3\ldots=i \\ k_1+k_2+k_3+\ldots=n}} \binom{n}{k_1 k_2 k_3 \ldots} \left( \alpha_1^{k_1} \alpha_2^{k_2} \alpha_3^{k_3} \ldots \right) \tag{29}$$

as shown in Eq. 5. Though this expression gives $f$ purely in terms of $\boldsymbol{\alpha}$, the presence of the conditional sum limits its practical utility compared to the iterative form.

## A.4 Proof of Theorem 2.3 (Empirical Rademacher complexity bound)

In this appendix we derive the bound on the empirical Rademacher complexity stated in Theorem 2.3 and show the consequent generalisation bounds. The early stages closely follow arguments made by Cortes et al. (2010). Recall that we have defined the hypothesis set

$$H = \{x \to \boldsymbol{w}^\top \boldsymbol{\psi}_{K_\alpha}(x) : |\alpha_i| \le \alpha_i^{(M)}, \|\boldsymbol{w}\|_2 \le 1\}, \tag{30}$$

with $\boldsymbol{\psi}_{K_\alpha} : x \to \mathbb{H}_{K_\alpha}$ the feature mapping from the input space to the reproducing kernel Hilbert space $\mathbb{H}_{K_\alpha}$ induced by the kernel $K_{\boldsymbol{\alpha}}^{\mathcal{G}}$.

The empirical Rademacher complexity $\widehat{\mathcal{R}}(H)$ for an arbitrary fixed sample $S = (x_i)_{i=1}^m$ is defined by

$$\widehat{\mathcal{R}}(H) := \frac{1}{m} \mathbb{E}_{\boldsymbol{\sigma}} \left[ \sup_{h \in H} \sum_{i=1}^m \sigma_i h(x_i) \right] \tag{31}$$

the expectation is taken over $\boldsymbol{\sigma} = (\sigma_1, \ldots, \sigma_m)$ with $\sigma_i \sim \text{Unif}(\pm 1)$ i.i.d. Rademacher random variables.

Begin by noting that

$$h(x) := \boldsymbol{w}^\top \boldsymbol{\psi}_{K_\alpha}(x) = \sum_{i=1}^m \beta_i K_\alpha(x_i, x) \tag{32}$$

where $\beta_i$ are the coordinates of the orthogonal projections of $\boldsymbol{w}$ on $\mathbb{H}_S = \text{span}(\boldsymbol{\psi}_{K_\alpha}(x_1), \ldots, \boldsymbol{\psi}_{K_\alpha}(x_m))$, where $\boldsymbol{\beta}^\top \mathbf{K}_\alpha \boldsymbol{\beta} \le 1$. Then we have that

$$\widehat{\mathcal{R}}(H) = \frac{1}{m} \mathbb{E}_{\boldsymbol{\sigma}} \left[ \sup_{\boldsymbol{\alpha}, \boldsymbol{\beta}} \boldsymbol{\sigma}^\top \mathbf{K}_\alpha \boldsymbol{\beta} \right]. \tag{33}$$

The supremum $\sup_{\boldsymbol{\beta}} \boldsymbol{\sigma}^\top \mathbf{K}_\alpha \boldsymbol{\beta}$ is reached when $\mathbf{K}_\alpha^{1/2} \boldsymbol{\beta}$ is collinear with $\mathbf{K}_\alpha^{1/2} \boldsymbol{\sigma}$, and making $\|\boldsymbol{\beta}\|_2$ as large as possible gives

$$\widehat{\mathcal{R}}(H) = \frac{1}{m} \mathbb{E}_{\boldsymbol{\sigma}} \left[ \sup_{\boldsymbol{\alpha}} \sqrt{\boldsymbol{\sigma}^\top \mathbf{K}_\alpha \boldsymbol{\sigma}} \right]. \tag{34}$$

Now note that

$$\boldsymbol{\sigma}^\top \mathbf{K}_\alpha \boldsymbol{\sigma} = \sum_{i=0}^{\infty} \alpha_i \boldsymbol{\sigma}^\top \mathbf{W}^i \boldsymbol{\sigma}. \tag{35}$$

$\boldsymbol{\sigma}^\top \mathbf{W}^i \boldsymbol{\sigma}$ may take either sign, and the sum is maximised by taking $\alpha_i = \alpha_i^{(M)}$ for positive terms and $\alpha_i = -\alpha_i^{(M)}$ for negative terms. Observe that

$$\sup_{\boldsymbol{\sigma}} \sup_{\boldsymbol{\alpha}} \sqrt{\boldsymbol{\sigma}^\top \mathbf{K}_\alpha \boldsymbol{\sigma}} = \sqrt{m \sum_{i=0}^{\infty} \alpha_i^{(M)} \rho(\mathbf{W})^i} \tag{36}$$

whereupon from Eq. 31 it follows that

$$\widehat{\mathcal{R}}(H) \le \sqrt{\frac{1}{m} \sum_{i=0}^{\infty} \alpha_i^{(M)} \rho(\mathbf{W})^i} \tag{37}$$

as stated in Thm 2.3. This bound is not tight for general graphs, but will be for specific examples: for example, when $\mathbf{W}$ is proportional to the identity so the only edges are self-loops. Nonetheless, it provides some intuition for how the learned kernel's complexity depends on $\mathbf{W}$.

As stated in the main text, Eq. 37 immediately yields generalisation bounds for learning kernels. Again closely following Cortes et al. (2010), consider the application of binary classification where nodes are assigned a label $y_i = \pm 1$. Denote by $R(h)$ the true generalisation error of $h \in H$,

$$R(h) = \Pr(yh(x) < 0). \tag{38}$$

Consider a training sample $S = ((x_i, y_i))_{i=1}^m$, and define the $\rho$-empirical margin loss by

$$\widehat{R}_\rho(h) := \frac{1}{m} \sum_{i=1}^m \min(1, [1 - y_i h(x_i)/\rho]_+) \tag{39}$$

where $\rho > 0$. For any $\delta > 0$, with probability at least $1 - \delta$, the following bound holds for any $h \in H$ (Bartlett and Mendelson, 2002; Koltchinskii and Panchenko, 2002):

$$R(h) \leq \widehat{R}_\rho(h) + \frac{2}{\rho} \widehat{\mathcal{R}}(H) + 3 \sqrt{\frac{\log \frac{2}{\delta}}{2m}}. \tag{40}$$

Inserting the bound on the empirical Rademacher complexity in Eq. 37, we immediately have that

$$R(h) \leq \widehat{R}_\rho(h) + \frac{2}{\rho} \sqrt{\frac{1}{m} \sum_{i=0}^\infty \alpha_i^{(M)} \rho(\mathbf{W})^i} + 3 \sqrt{\frac{\log \frac{2}{\delta}}{2m}} \tag{41}$$

which shows how the generalisation error can be controlled via $\boldsymbol{\alpha}^{(M)}$ or $\rho(\mathbf{W})$.

## A.5 Learning asymmetric $f^{(N)}$

Recalling from Eq. 1 that the modulation functions $(f_1, f_2)$ do not necessarily need to be equal for unbiased estimation of some kernel $\mathbf{K}_{\boldsymbol{\alpha}}$, a natural extension of Sec. 3.4 is to introduce *two* separate neural modulation functions $f_{1,2}^{(N)}$ and train them both following the scheme in Sec. 3.4. Intriguingly, even with an initialisation where $f_2^{(N)}$ encodes 'lazy' behaviour (deposits almost all its load at the starting node – see Sec. 2) and $f_1^{(N)}$ is flat, upon training the neural modulation functions quickly become very similar (though not identical). See Fig. 6. We use the same optimisation hyperparameters and network architectures as in Sec. 3.4. Rigorously proving the best possible choice of $(f_1, f_2)$, including whether e.g. a symmetric pair is optimal, is left as an exciting open theoretical prob-

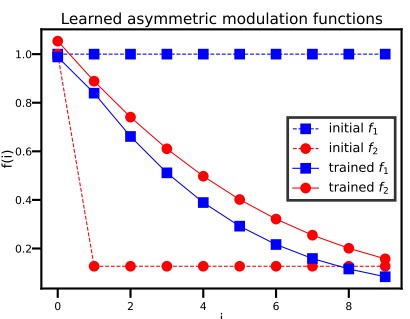

Figure 6: Modulation functions $(f_1, f_2)$ parameterised by separate neural networks before and after training to target the 2-regularised Laplacian kernel.

lem. We note that, whilst parameterising two separate neural modulation functions gives a more general mechanism (with the symmetric pair as a special case), it also doubles the number of parameters required so slows training and evaluation.

## A.6 Further approximation error results for Erdős-Rényi graphs

Here we further investigate the behaviour of the g-GRFs kernel approximation as the properties of the graph change. To do this, we generate random graphs using the Erdős-Rényi model, where each of the $\binom{N}{2}$ possible edges are present with probability $p_{\text{edge}}$ or absent with probability $1 - p_{\text{edge}}$. Edges are independent.

Firstly, we investigate how the approximation error varies with graph sparsity. We take a fixed number of nodes ($N = 100$) and control the sparsity by varying $p_{\text{edge}}$ between 0.1 and 0.9. We then approximate the diffusion kernel using g-GRFs with a termination probability $p_{\text{halt}} = 0.1$ and 8 walkers. For each graph we compute the relative Frobenius norm error between the true ($\mathbf{K}$) and approximated ($\widehat{\mathbf{K}}$) kernels: namely, $\|\mathbf{K} - \widehat{\mathbf{K}}\|_F / \|\mathbf{K}\|_F$. We repeat 100 times to obtain the standard deviation of the mean error estimate. Fig. 7 shows the results; approximation quality degrades slightly as $p_{\text{edge}}$ grows, but remains very good throughout (error $< 0.00275$).

Next, we investigate the scalability of the method by comparing the time for exact and approximate evaluation of the diffusion kernel as the size of the graph grows. We fix $p_{\text{edge}} = 0.5$ and vary the number of nodes $N$ between 100 and 12800. For every graph, we measure the wall-clock time for i) exact computation of $\mathbf{K}$ by computing the matrix exponential and ii) approximate computation of $\widehat{\mathbf{K}}$ using g-GRFs (8 walkers, $p_{\text{halt}} = 0.5$, regulariser $\sigma = 0.5$). Fig. 8 shows the results. Naturally, the exact method scales worse, becoming slower for graphs bigger than a few thousand nodes, and by the largest graph g-GRFs are already faster by a factor of 7.8. We also measure the relative Frobenius norm between the true and approximated Gram matrices to check that the quality of estimation remains good. This is shown by the green line, which indeed remains almost constant and takes very small values for the error ($\simeq 0.005$).

Figure 7: Approximation error vs. edge-generation probability for the diffusion kernel on a graph of size $N = 100$. The quality remains good as sparsity changes, varying within a narrow range.

Figure 8: Wall clock time for exact and approximate kernel evaluation as the number of nodes varies. g-GRFs scale better and are faster with a few thousand nodes. The approximation error remains low as $N$ grows.

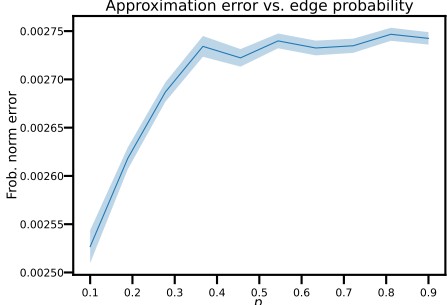 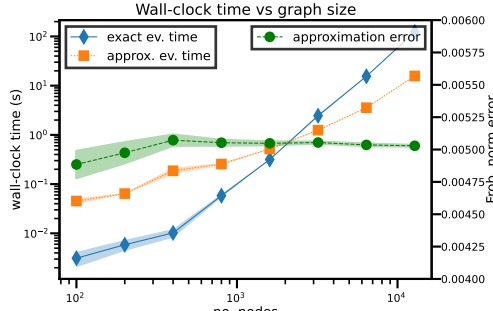

### A.7 FURTHER EXPERIMENTAL DETAILS

In this short section, we provide further experimental details and discussion to supplement the main text.

1. **Choice of $p_{\textbf{halt}}$**: The termination probability encodes a simple trade-off between approximation quality and speed; if $p_{\text{halt}}$ is lower, walks tend to last longer and sample more subwalks so give a better approximation of graph kernels. In practice, any reasonably small value works well, as also reported for the original GRFs mechanism (Choromanski, 2023). In experiments we typically choose $p_{\text{halt}} \sim 0.1$.

2. **Choice of $\sigma$ and kernel convergence**: After Eq. 2 we noted that, when approximating some fixed kernel $\mathbf{K}_{\boldsymbol{\alpha}}(\mathbf{W}) = \sum_{k=0}^{\infty} \alpha_k \mathbf{W}^k$, we assume that the sum does not diverge. It would not be possible to construct a finite random feature estimator if this were not the case. We require that $\sum_{k=0}^{\infty} \alpha_k \rho(\mathbf{W})^k$ is finite, with $\rho(\mathbf{W}) \coloneqq \max_{\lambda \in \Lambda(\mathbf{W})} (|\lambda|)$ the spectral radius of the weighted adjacency matrix ($\Lambda(\mathbf{W})$ is the set of eigenvalues of $\mathbf{W}$). When considering e.g. the diffusion kernel in the literature, this is ensured by multiplying $\mathbf{W}$ by some regulariser $\sigma \in \mathbb{R}_+$, tak-

ing e.g. $\mathbf{W} \to \sigma\mathbf{W}$ whereupon $\lambda_i \to \sigma\lambda_i \; \forall \; i = 1, ..., N$. This is the reason for extra parameters $\{\sigma, \alpha\}$ in the kernel expressions in Table 1. As we noted in Sec. 3.5, this is exactly equivalent to transforming the modulation function $f(i) \to f(i)\sigma^i \; \forall \; i$. Where space allows, we report the exact choice of $\sigma$ with each experiment (though empirically it does not modify our conclusions). In Sec. 3.5 we take the weighted adjacency matrix $\mathbf{W} := [a_{ij}/\sqrt{d_i d_j}]_{i,j=1}^N$ with $a_{ij} = 1$ if node $i$ is connected to $j$ and 0 otherwise, and $d_i$ the degree of node $i$. We then regularise by taking $\mathbf{W} \to \sigma\mathbf{W}$ with $\sigma = 0.025$, which is small enough for convergence even with the largest graph considered. We use $m = 16$ walkers and $p_{\text{halt}} = 0.5$.

