# OpenReview forum: "General Graph Random Features"
_ICLR.cc/2024/Conference — ICLR 2024 poster_

### Official Review · Reviewer_TqUg · 2023-10-27

**Soundness:** 3 good
**Presentation:** 3 good
**Contribution:** 2 fair
**Rating:** 8
**Confidence:** 3

**Summary:**

The paper introduces techniques to approximate a family of kernels for comparing the nodes of a graph. The methods rely on modulation functions that are either analytically derived from the original kernel, or learned from data by optimizing the downstream task.
Experiments show that as the representation grows the performance improves and that the learned kernels perform well in inductive tasks too.

**Strengths:**

- The proposed method allows approximating the entire family of kernels based on weighted adjacency matrices.
- Learning the modulation function shows how to successfully retrieve effective kernels within the considered family.
- Examples show how to apply the proposed methods in different scenarios and tasks.
- The paper reads well, and the contribution w.r.t. prior work is fair and clearly stated.

**Weaknesses:**

- The proposed method is an extension of previous works, in particular, Choromansky (2023).
- The experiments do not clearly show the price that one has to pay in terms of compute time when requesting better accuracy. Experiments show only relations between the number of walks and the accuracy/error. Linking accuracy and execution time (e.g., in standard machines) would have given a better sense of the relevance of the proposed methods.
- The kernels apply only to weighted graphs without node and edge attributes.
- The use of the term 'universal' is therefore misleading; as the authors say, the provided random features can approximate only kernels from a specific family.

**Questions:**

- Alg. 1: shouldn't the normalization be $1/\sqrt{m}$ instead of $1/m$?
- Fig. 3: please specify if the extremely narrow shaded area around the lines refers to +/- one standard deviation.
- Sec. 3.4: it is claimed that the modulation function learned on one data set can be effectively applied to other graphs, referring to Tab. 3. However, I don't see specified how the table has been generated. Please provide more details.
- Tab. 3 and 4: I suggest introducing the notation used for the standard deviation.
- I suggest discussing the impact of the fact that the sum in Eq. 2 is not always well-defined - an aspect that is only briefly mentioned in the paper.
- I suggest discussing the pros and cons of symmetric vs asymmetric approaches.

---

> ### Author Response · Authors · 2023-11-12
> **Rebuttal -- thank you for the review**
>
> We thank the reviewer for reading the manuscript and are pleased that they report that the paper reads well. We address their questions and points of minor misunderstanding below.
>
> 1. **Difference from Choromanski (2023)**: The reviewer correctly notes that this work extends Choromanski's paper (https://arxiv.org/abs/2305.00156, ICML 2023, oral presentation). Whilst this previous work presents a random feature mechanism for unbiased approximation of the $d$-regularised Laplace kernel, our extension allows unbiased approximation of *any* function of a weighted adjacency matrix. It is therefore significantly more general; it contains Choromanski's scheme as a special case. Moreover, unlike its predecessor, our method also allows for implicit kernel learning via a neural modulation function. Hence, our work is substantially broader in scope.
> 2. **Accuracy vs compute**: Our algorithm has the convenient feature that walkers are independent and straightforwardly distributed across machines. This means that the run time to construct features is directly proportional to the number of walkers, so the plots in e.g. Fig 2 would look identical if the $x$-axis variable were replaced with simulation time. Moreover, from Alg. 1 it is clear that the time-complexity to construct features is subquadratic so scales better than exact methods (cubic). That said, we agree that adding some wall-clock times to compare the exact and approximated speeds might be interesting for practitioners. We are measuring these and will add the results to the appendix.
> 3. **Classes of kernels being approximated**: The reviewer is correct that our algorithm can approximate arbitrary functions of a weighted adjacency matrix, which is not the same as any kernel. This is the sense in which they are 'universal', as we state in the first sentence of the abstract. We chose this name to disambiguate from 'graph random features' (https://arxiv.org/abs/2305.00156, ICML 2023) which can only approximate the $d$-regularised Laplacian function. We intended to be unambiguous and welcome suggestions to make our contribution clearer -- does the reviewer think 'general graph random features' would avoid confusion? We would be happy to make any reasonable changes to the name.
> 4. **Normalisation factor**: the $\frac{1}{m}$ normalisation factor is actually correct since we are interested in the distribution over subwalks of the average walk. We invite the reviewer to read Sec. A.2 for more detail. This might seem strange compared to e.g. random Fourier features (https://people.eecs.berkeley.edu/~brecht/papers/07.rah.rec.nips.pdf, NIPS 2007) which are normalised by the square root of the feature length, but upon careful inspection this is indeed the right factor.
> 5. **Shaded area and standard deviation**: The shaded area is indeed the standard deviation on the estimate of the mean, i.e. the standard deviation of one trial divided by the square root of the number of trials. We will clarify this in the manuscript. We will also clarify bracket notation for standard deviation on the last digit.
> 6. **Table 3 and kernel learning**: To produce Table 3, the modulation function was trained for better quality approximation on the smallest graph ('on the small Erdős-Rényi graph ($N = 20$) with $m = 16$ walks, we minimise the loss'). We then freeze the modulation function and test the quality of kernel approximation on the other bigger graphs. Even though $f$ was only trained on to be optimal on the smallest graph (top row of table), it also improves approximation quality for all the remaining graphs (remaining rows), even when $N$ is bigger by a factor of $60$. We will amend the text to make this important point clearer.
> 7. **When the kernel is not defined**: The reviewer is correct to observe that if the kernel diverges then we cannot hope to approximate it using finite random features. In the literature, this is avoided by multiplying the weighted adjacency matrix by a regulariser $\sigma^2$ which downscales its spectral radius. We will add further discussion to the manuscript.
> 8. **Symmetric vs asymmetric approaches**: We sincerely thank the reviewer for their interesting question. Eq. 4 encodes is the condition for an unbiased estimator; symmetric functions is the special case when we constrain $f_1 = f_2$. This has the benefit of requiring half as many parameters in the neural modulation functions, and empirically during training functions tend to become close to symmetric anyway (see Fig. 6). A detailed study of the effect of different choices among the possible modulation function pairs, including whether the symmetric case is best, is an open and exciting research direction. We will add this discussion and look forward to tackling these questions fully in the future.
>
> We again warmly thank the reviewer and invite any further questions. We hope that, if satisfied, they will consider raising their score.

---

> ### Author Response · Authors · 2023-11-13
> **Updated pdfs**
>
> The reviewer might be interested to see the updated main and supplementary pdfs. Changes are indicated in red. Prompted by their helpful suggestion, we have now added a further experiment in Sec. A.6 plotting the wall-clock times for exact and u-GRF kernel evaluation as the number of nodes grows. For this experiment, exact methods become slower for graphs with more than $\sim 1600$ nodes, and by the time we reach the biggest graph considered ($12800$ nodes) our algorithm is already about $8$ times faster. We also plot the kernel approximation error as $N$ grows to check that the quality of the estimate is not degrading. We hope this clarifies the rough speedups that can be expected from using u-GRFs in different regimes. We have also explained the standard deviation shading and notation, added comments on regularisation and the requirement that the target kernel does not diverge (Sec. A.7), and added discussion of the strengths and weaknesses of symmetric and asymmetric modulation functions. We thank the reviewer again for suggesting these improvements.

---

> > ### Comment · Reviewer_TqUg · 2023-11-17
> >
> > I thank the authors for having addressed all my comments and questions and I am happy to increase my score.
> >
> > However, I would like to mention that I still believe that the term "universal" is ambiguous in this context and I suggest using the term "general", as suggested by the authors, in both the title and the text.

---

> > > ### Author Response · Authors · 2023-11-17
> > > **Thanks**
> > >
> > > Thanks for the reply and for raising the score. We agree that 'general' might avoid ambiguity and will update the manuscript. We thank the reviewer again for their time and thoughtful feedback.

---

### Official Review · Reviewer_9UzR · 2023-10-27

**Soundness:** 2 fair
**Presentation:** 3 good
**Contribution:** 3 good
**Rating:** 8
**Confidence:** 4

**Summary:**

A large span of interesting problems involving graph data require to compute a function of the weighted adjacency matrix. In most cases, using a naive procedure to compute this function scales (at least) cubicly with the number of nodes in the graph.

The authors propose a new algorithm to circumvent this computational limitation. Inspired by a recent work by Krzysztof Marcin Choromanski (KMC), the authors introduce a new algorithm to approximate the value of arbitrary functions of the weighted matrix of the graph by using graph random features. Graph random features rely on the possibility to write the function of the weighted adjacency matrix as the expectation of finite dimensional vectors. The former vectors are obtained by using random walks on the graph. Contrary to the KMC's work, the author introduce the notion of modulation function that controls each walker’s load as it traverses the graph. This extra modulation function allows the algorithm to approximate arbitrary functions of the weighted adjacency matrix.

The authors also show that this modulation function can be learned using neural network to solve some given task of interest.

**Strengths:**

- The paper brings two main contributions. First the introduction of the notion of modulation function and its use in a random walk algorithm to estimate arbitrary functions of the weighted adjacency matrix of a graph. Second, the possibility to learn using neural network this modularity function to automatically selects a graph representation well suited for some downstream task.

Therefore, this work not only allows to reduce the computational burden to solve some important graph related questions using graph random features and the proposed algorithm, but the method can be used to automatically find the relevant graph representation for a task of interest.

- The paper includes both interesting theoretical results and several numerical experiments (such as node attribute prediction on triangular mesh graphs).

- The paper is well written and the authors pan to make their code publicly available in case their work is accepted.

**Weaknesses:**

- The authors highlight several times in the paper that one major issue regarding in terms of computational complexity is to inverse the matrix K(W) (where W is the weighted adjacency matrix of the graph). However, in the numerical experiments presented in the paper, the authors do not propose an example of application of their method to inverse such matrix K. It might be relevant to add such experiment in the paper or at least to explain how the method can be used to efficiently compute the inverse.

- My other comments or suggestions for improvements are mainly related to the experiment section and I refer to the list of my questions.

**Questions:**

I thank the authors for their nice work. Here are my questions and comments.

- The authors did a good job in providing several experiments to illustrate their method. Nevertheless, I think it would have been good to propose an experiment on real data that tackles one of the main applications mentioned in the introduction (such as community detection or recommender systems).

In particular, it is still an ongoing research question to find the right representation of a graph to use spectral methods for community detection. Indeed, using the adjacency matrix is known to be suboptimal in particular when dealing with sparse graphs. That's why people have proposed to use the non-backtracking matrix or the Bethe-Hessian. Since the authors show the interesting possibility to learn the modularity function to automatically selects a graph representation well suited for some downstream task, I think that a very interesting experiment would be to use their method to automatically select a good kernel for community detection. In particular, does the algorithm converge to a representation close to the non backtracking matrix ?


- The authors do not discuss really how some parameters of the method are chosen in practice (such as the parameter $p_{halt}$), and how these choices influence the performance of the algorithm.

- Real world graphs are known to be sparse. It would be interesting to comment of the influence on the algorithm when dealing with sparse graphs. A possibility would be to conduct simulated experiments by varying the sparsity level of the graph and see the impact of the performance.

Here are some typos and additional comments:

- In Eq.(9), I think a notation is missing since $\alpha$ is not appearing in the mapping $x\mapsto w^{\top} \psi_{K}(x)$. I think the correction would be $x\mapsto w^{\top} \psi_{K_{\alpha}}(x)$.

- At the first line in section 3.1, I think a "for" is missing. It should be "do indeed give unbiased estimates for the graph kernels".

- Between Eq.(13) and Eq.(14), I think there is an issue with the notations. Should $\Phi$ depend on $\tau_j$ ? Based on what I could understand, $\Phi$ is a rectangular matrix of size $n\times N$ where the $j$-th row is $\Phi_j = ((t-\tau_j) \phi(i))_{i=1}^N$.

---

> ### Author Response · Authors · 2023-11-12
> **Rebuttal -- thank you for the review**
>
> We thank the reviewer for their reading of the manuscript and thoughtful comments. We are pleased that they identify the interesting theoretical and numerical results and clear writing. We address all questions and points of concern below.
>
> 1. **Efficiency gains and the kernel inverse**: The reviewer correctly notes that the large cost of kernel inversion is a big motivating factor for constructing random feature mechanisms. In particular, in the presence of explicit random features for a kernel $K$ so that $K \simeq \phi^\top \phi$, we can avoid needing to compute its inverse $K^{-1}$. This is well-established in the random features literature. However, graph kernel methods have the additional challenge that even *computing* $K$ scales cubically with the number of nodes $N$. Our predominant focus is on overcoming this (using our sub-quadratic algorithm), with the understanding that the other benefits of explicitly-manifested features also follow. We will make this clearer in the manuscript.
>
> 2. **Bethe Hessian and community detection**: The reviewer rightly identifies one of the novel contributions of the paper as a scalable new technique for implicit kernel learning in random feature space. Though this only constitutes one short section (the paper also introduces an unbiased mechanism for kernel estimation and learns the modulation function to improve kernel approximation quality), we sincerely thank them for their interest in this exciting research direction. We agree that there is plenty of scope for interesting work, including their excellent suggestion to apply it to community detection problems and see whether the implicitly learned kernel approximates any popular choices from the literature. We are excited to explore this aspect to u-GRFs further, but for now must defer it as future work.
>
> 3. **Choice of parameters**: When approximating a fixed graph kernel, the only free choice of parameter is $p_\text{halt}$. It encodes a simple trade-off between speed and approximation quality; if $p_\text{halt}$ is lower, walks last longer and sample more subwalks so give a better approximation of graph kernels. In practice, any small value works well (as also reported in https://arxiv.org/abs/2305.00156 which introduced the GRF mechanism -- ICML 2023) so we typically take $p_\text{halt} \sim 0.1$. We will add this discussion to the manuscript.
>
> 4. **Sparsity and performance**: We thank the reviewer for their interesting suggestions. We have included experiments on both synthetic and real-world graphs which naturally have a range of sparsities, and in all cases the performance is similar (see e.g. the plots in Fig. 2 or the clustering errors in Table 2 -- they do not change much between graphs). That said, we agree that it would be interesting to systematically investigate how approximation error changes with sparsity so we are running this extra experiment and will add the results to the Appendix.
>
> 5. **Additional comments and typos**: We thank the reviewer for their observation about Eq. 9 and agree that their suggestion would make the notation clearer, so we have added an $\alpha$ subscript. We have also added the word 'for' to the sentence at the start of Sec. 3.1. In Eq. 14, $\boldsymbol{\phi}_j$ actually refers to the $N \times N$ random feature matrix used to approximate the particular time evolution operator $\exp(\mathbf{W}(t - \tau_j))$ (i.e. the diffusion kernel corresponding to one particular timestep indexed by $j$) rather than $j$ indexing a row. We will clarify this notation and thank the reviewer for flagging this.
>
> We again thank the reviewer for their careful reading of the manuscript and invite them to respond with any further questions. If satisfied, we hope they will consider raising their score.

---

> ### Author Response · Authors · 2023-11-13
> **Updated pdfs**
>
> To provide a further update, we have now uploaded new versions of the main and supplementary pdfs. Changes and additions are indicated in red. Prompted by the reviewer's interesting suggestion, we have included an additional experiment in the appendix (Sec. A.6) investigating the effect of sparsity on the quality of kernel approximation. We fix the number of nodes of an Erdős-Rényi graph and vary the edge-generation probability from 0.1 to 0.9, and measure the relative Frobenius norm error between the true and approximated Gram matrices with the diffusion kernel in every case. The error actually remains very small throughout, but does increase slightly as $p_\text{edge}$ grows. Indeed, it is smallest for sparse graphs which the reviewer rightly identifies as corresponding to 'real data'. We have also added a section (A.7) to discuss the choice of parameters and corrected typos and unclear notation. We thank the reviewer for prompting these additions and improvements.

---

> > ### Comment · Reviewer_9UzR · 2023-11-22
> >
> > I thank the authors for their response to my initial review of the manuscript. I appreciate the time and effort invested in addressing the concerns raised during the review process.
> > Based on the improvements made and the addressed concerns, I have decided to raise my rating for the manuscript.

---

> > > ### Author Response · Authors · 2023-11-22
> > > **Thanks again**
> > >
> > > We thank the reviewer again for their time and thoughtful feedback. They have helped us improve the manuscript.

---

### Official Review · Reviewer_jWMX · 2023-10-28

**Soundness:** 3 good
**Presentation:** 3 good
**Contribution:** 3 good
**Rating:** 8
**Confidence:** 4

**Summary:**

This paper introduces universal graph random features (u-GRFs) based on random walks in the graph. The authors show that u-GRFs can be used to unbiasedly approximate functions of the weighted adjacency matrix of a graph, which include popular graph kernels such as d-regularized Laplacian kernel, p-step random walk kernel and the heat diffusion kernel. In addition, the authors extend their algorithm for estimating fixed graph kernels to a learnable approach, parametrizing a fixed modulation function with a two-layer neural network. The authors provide a simple generalization bound for the corresponding class of learnable graph kernels. Finally, extensive experiments over various application settings are provided to demonstrate the good performance and potentially wide applicability of u-GRFs.

**Strengths:**

- This paper extends the recent work of Choromanski (2023) on random walk-based graph random features. Compared to the previous work, this paper has two notable innovations. The first is an introduction of a modulation function which enables estimation of arbitrary functions of the weighted adjacency matrix, the second is an extension to a parametrized setting with a generalization bound. While this work is not the first to study graph random features, it has an easy-to-identify contribution which I think makes it a nice addition to the community.

- The overall presentation and writing are clear and easy to follow. I enjoyed reading this paper.

- The experiments, though limited to a few small graphs, are reasonably extensive and showcase the potentially wide applicability of the proposed universal graph random features.

**Weaknesses:**

- The experiments are carried over small graphs. It would be nice to include some experiments on larger graphs, e.g. N=10,000. For example, fix the Erdos-Renyi model but vary the number of nodes. I wonder if and how the approximation error would scale with N, while keeping everything else fixed.

- For experiments on node clustering, Table 2 shows the difference between u-GRFs and the exact kernel. It would be nice to have another table that shows the actual clustering accuracy when compared against the ground-truth information, given the ground-truth number of clusters in each graph.

**Questions:**

- Given Theorem 2.1 and Equation 4, as long as f_1 and f_2 satisfy Equation 4 we will have an unbiased estimator. In that case, what is particularly interesting/special about symmetric modulation functions? Does using symmetric modulation functions reduce estimator variance (either empirically or theoretically)?

- Why did you use the pairwise metric in Equation 15 as opposed to node-wise accuracy compared with the clustering result using the exact kernel?

- If I understood correctly, the numbers in the parenthesis in Table 3 and Table 4 are standard deviations. Why are they integers?

---

> ### Author Response · Authors · 2023-11-12
> **Rebuttal -- thank you for the review**
>
> We thank the reviewer for their reading of the manuscript and detailed feedback. We are happy that they note the clarity of the writing and wide applicability of u-GRFs. We answer their interesting questions below.
>
> 1. **Graph size**: We consider graph node clustering with $N=3300$ and node attribute prediction with $N=4350$ which are already sufficiently large that exact methods, penalised by cubic time-complexity scaling, become slow (for clustering, by a factor of $> 5$ compared to u-GRFs). Nonetheless, we agree that it would be interesting to continue to push the methods to yet larger graphs. **Therefore, following the reviewer's suggestion, we also ran the node attribute prediction experiment on the 'cycloidal' mesh which has $21384$ nodes**. In this regime exact kernel evaluation is extremely slow, but once again u-GRFs perform well and the learned kernel (trained on the cylinder graph) does best. Specifically, the normalised differences in prediction error compared to the learned kernel are: $0.065(4)$, $0.129(6)$ and $0.011(1)$ for the $1$-regularised Laplacian, $2$-regularised Laplacian and diffusion kernels respectively. Indeed, throughout the work we see that performance is similar across graph size (see e.g. Fig. 2, where the plots are essentially identical even as $N$ differs by a factor of $60$). However, we will also add a plot to the appendix explicitly comparing number of $N$ and the kernel approximation error; we thank the reviewer for their excellent suggestion.
>
> 2. **Clustering experiment and metric**: We are unsure what the reviewer means by 'groundtruth clusters' for these graphs, but will be happy to respond if they clarify. The pairwise clustering metric in Eq. 15 is a standard way to capture the quality of clustering (the accuracy becomes low if lots of pairs that should be in the same cluster are in different clusters or vice versa), but our scheme would perform well under any reasonable way of comparing the clusterings with the exact and approximated kernels.
>
> 3. **Modulation function symmetry**: The reviewer is correct that Eq. 4 is sufficient for an unbiased estimator of a graph kernel, so $f_1$ and $f_2$ need not in general be equal. The symmetric special case is useful because it needs half the number of learnable parameters  compared to the general case when we use neural modulation functions. Moreover, modulation functions initialised differently empirically tend to converge towards being close to symmetric anyway (see Fig. 6 in Sec. A5), suggesting that these give lower-variance estimates. The reviewer is right to identify this as an exciting and challenging direction for future theoretical work; we thank them for their interest.
>
> 4. **Standard deviation notation**: the bracketed number gives the standard deviation of the final digit of a long number more compactly, e.g. $0.123(4)$ means $0.123 \pm 0.004$. We will clarify this notation in the manuscript.
>
> We again thank the reviewer for their time and detailed review. We welcome any further questions.

---

> ### Author Response · Authors · 2023-11-13
> **Updated pdfs**
>
> As a further update, we have now uploaded new versions of the main and supplementary pdfs. Additions and changes are indicated in red. The reviewer might be particularly interested to see Sec. A.6 of the Appendix, which includes a new plot of wall-clock time and approximation error vs the number of nodes in an Erdős-Rényi graph. As suggested, we scale graphs to as big as 12800 nodes where exact computation becomes prohibitively slow. Our method continues to perform well in this regime, with the relative Frobenius norm error between the true and approximated Gram matrices remaining very small. We have also added discussion comparing symmetric and asymmetric modulation functions and clarified the standard deviation notation. We warmly thank the reviewer for suggesting these interesting additions and important clarifications.

---

> > ### Comment · Reviewer_jWMX · 2023-11-22
> >
> > I thank the authors for their responses and their efforts to incorporate additional evaluations/discussions in the paper. The clarity of the paper has been improved. This reinforces my opinion that this is a high-quality contribution.

---

### Official Review · Reviewer_44jF · 2023-11-06

**Soundness:** 3 good
**Presentation:** 3 good
**Contribution:** 3 good
**Rating:** 8
**Confidence:** 5

**Summary:**

This paper extends a result of Choromanski (2023) in order to compute efficiently an approximation of any functions of a weighted adjacency matrix. The main idea is to add a so-called modulation function in an algorithm based on random walks in order to compute the feature projector.

###
After a discussion with the authors, I changed my ratings about soundness and contribution and my general rating as the authors clarified my mistake about the proof of their Theorem.

**Strengths:**

This result is a nice extension of the previous result due to Choromanski (2023) allowing us to compute approximations of a variety of graph kernels.

**Weaknesses:**

I do not understand the proof of the main Theorem 2.1 see below.

The experiments could be improved. The authors mainly compare the results given by their algorithm to the real kernel, i.e., measure the approximation error due to their algorithm. It would be more convincing to have used their algorithm on a 'real' graph problem with a large number of nodes and where standard kernel methods are not possible.

**Questions:**

I think equation (19) in the appendix is not correct. This makes the proof of the main result Th 2.1 incorrect. Equation (19) involves only the modulation function evaluated on the total length of the walk, but in Algorithm 1 on line 8, the update for the feature vector uses all values of the modulation function evaluated at $k\leq$ length(walk).
You probably mean in the indicator function that $\omega_{iv}$ is a 'prefix' of the total walk?
But then, when you consider the full walk, you need to have a factor $p$ and not only $(1-p)^len$?

I find the term universal quite misleading. You are approximating arbitrary functions of the weighted adjacency matrix, but these are not all possible graph kernels. Indeed, it would be nice to have the expressive power of the kernel you are approximating if possible.

---

> ### Author Response · Authors · 2023-11-12
> **Rebuttal -- thank you for the review**
>
> We thank the reviewer for reading the manuscript. We address questions, concerns and points of minor misunderstanding below.
>
> 1. **Proof clarification**: We are grateful that the reviewer checked the supplementary material, but we respectfully maintain that there is no mistake: Eq. 19 is in fact correct and the theorem follows. We invite the reviewer to consider the paragraph directly after Eq. 19 -- in particular, '$\mathbb{I}$ is the indicator function which evaluates to 1 if its argument is true (namely, $\omega_{iv}$ is a subwalk of $\bar{\Omega}_k^{(i)}$, the $k$-th walk sampled out of [node] $i$), and $0$ otherwise'. To be clear, $\omega_i$$_v \in$ $\bar{\Omega}_k^{(i)}$ is the event that $\omega_i$$_v$ is a 'prefix' of $\bar{\Omega}_k^{(i)}$, which has probability: $p(\omega) $$= (1-p)^{\text{len}(\omega)}\prod_1^{\text{len}(\omega)} \frac{1}{d_i}$ (we dropped the $iv$ subscript because of markdown rendering errors and the product is over $i$). This expression does not need an extra factor of $p$ because we just require that the walk we sample *includes* $\omega_i$$_v$ as a subwalk, not that it terminates at $v$ -- the walk is free to continue afterwards. Note then that, for the $k$th walk sampled out of node $i$ (denoted $\bar{\Omega}_k^{(i)}$), considering all the $v$ coordinates of the random feature, we get contributions to the vector $\phi(i)$ for every walk $\omega_i$$_v$ that is a prefix of $\bar{\Omega}_k^{(i)}$. This happens $\text{len}(\bar{\Omega}_k^{(i)})$+1 times (all the prefix walks, including the walk of length $0$) at different coordinates $v$, which corresponds exactly to the updates at every timestep the reviewer identifies in line 8 of Alg. 1. The crux here is the relationship between $\omega_i$$_v$ (the variable in the sum over all the possible walks between $i$ and $v$ on the graph) and $\bar{\Omega}_k^{(i)}$ (a particular finite-length walk we have sampled out of $i$) via the event  $\omega_i$$_v \in \bar{\Omega}_k^{(i)}$ ($\omega_i$$_v$ is a prefix subwalk of $\bar{\Omega}_k^{(i)}$). On careful inspection, Eq. 19 then corresponds exactly to the estimator generated by the algorithm. We invite the reviewer to respond with further questions.
> 2. **Experiments**: The reviewer is correct that many of our experiments compare results with the approximated kernel to the groundtruth. This is because our central contribution is a novel algorithm for unbiased and efficient approximation of arbitrary functions of a weighted adjacency matrix. A natural way to check whether the approximation is good by comparing the exact and approximated results. This includes on big 'real' graphs where exact methods are already slow due to the cubic-time complexity scaling (e.g. 'citeseer' has $3300$ nodes and 'torus' has $4350$ nodes for the clustering and kernel regression experiments respectively). Here, the speedup from using u-GRFs is already a factor of $> 5$. **Following the reviewer's comment we ran the node attribute prediction task again on the 'cycloidal' graph which has $21384$ nodes**. The kernel we trained on the small 'cylinder' graph again performed best, with normalised differences in prediction error of: $0.065(4)$, $0.129(6)$ and $0.011(1)$ for the $1$-regularised Laplacian, $2$-regularised Laplacian and diffusion kernels respectively. **This is a very big graph for which exact kernel computation is prohibitively slow, but our scaleable u-GRF method continues to perform well**.
>
> 3. **'Universal' graph random features**: We added the word 'universal' to disambiguate from 'graph random features' (https://arxiv.org/abs/2305.00156, ICML 2023, oral presentation) which can only approximate the $d$-regularised Laplace function. In the first sentence of the abstract we state that this refers to the ability to approximate arbitrary functions of a weighted adjacency matrix. In the second sentence we clarify that this includes 'many' (not all) popular graph node kernels. We intended to be unambiguous about this fact and strongly agree that it is important to be clear about contributions, so we welcome feedback and suggestions for how to make this more obvious. Does the reviewer think that e.g. 'general graph random features' would fit better? We are happy to rename them to any sensible suggestion.
>
> We thank the reviewer again and invite them to respond with further queries. We hope the clarifications will be sufficient for them to consider raising their score.

---

> > ### Author Response · Authors · 2023-11-13
> > **Updated pdfs**
> >
> > We invite the reviewer to inspect the updated main and supplementary pdfs, with changes indicated in red. They might be particularly interested in the new experiments, including on bigger graphs ($>10000$ nodes) where exact kernel evaluation becomes extremely slow. We look forward to their response and will be happy to answer any further questions.

---

> > > ### Author Response · Authors · 2023-11-16
> > > **Anything else we can clarify?**
> > >
> > > Could the reviewer kindly confirm that we have made clear why Eq. 19 is correct and that the proof follows? If satisfied, would they please consider raising their score, or else let us know what remains unresolved?

---

> > > > ### Comment · Reviewer_44jF · 2023-11-16
> > > > **Agree with equation 19**
> > > >
> > > > Thanks for the clarification. You are using $\in$ with two different meanings in $\sum_{\omega_{iv}\in \Omega_{iv}}$ where this is the standard meaning for an element being in a set and in the indicator function where $\in$ means being a prefix. You should probably clarify this.
> > > >
> > > > Regarding the title, I tend to prefer "general graph random features". Reading "universal" in the title, I thought about approximation theory which is not the topic of the paper.

---

> > > > > ### Author Response · Authors · 2023-11-16
> > > > > **Thanks**
> > > > >
> > > > > Thanks for the response. We agree on both points and will update the manuscript. We thank the reviewer for their thoughtful feedback.

---

### Meta-Review · Area_Chair_vxTL · 2023-12-09

**Metareview:**

**Summary**
This paper introduces a random walk-based algorithm that provides an unbiased estimation of functions of a weighted adjacency matrix for a given graph. Unlike the naïve treatment of the Gram matrix, which has cubic time complexity with respect to the number of nodes, the proposed algorithm successfully reduces it to subquadratic time complexity. This reduction is achieved through random walk-based estimation and a newly proposed modulation function that controls contributions from various lengths of walks. The effectiveness of the proposal has been demonstrate thorough experiments.

**Strengths**
- The contribution of this paper is twofold: theoretical and practical. In both aspects, the paper presents promising results, which have also been appreciated by the reviewers.
- This paper is well-written. Presentation is clear and the paper is easy to follow.

**Weaknesses**
- The novelty is somewhat weak. The main algorithm is a simple modification of the GRFs proposed by Choromanski (2023).
- I wonder if the technique of using the modularity function is similar to the well-known geometric random walk kernel. Since its limitations have been theoretically analyzed (see [1]), a discussion about the relationship to such work would be interesting.
- Additionally, while this paper focuses on kernels for nodes within a single graph, considering the product graph of a pair of graphs could extend the discussion to kernels between a pair of graphs rather than individual nodes. It would be interesting to explore such an extension in future work.

[1] Sugiyama, M., Borgwardt, K.M., Halting in Random Walk Kernels, NIPS2015.

**Justification For Why Not Higher Score:**

As I have mentioned in the meta-review, the novelty of this paper is not particularly strong, as a substantial part of the main algorithm is derived from previous literature, and similar concepts are known about the new component, the modulation function. Taking into account my opinion and other concerns raised by the reviewers, I believe "Accept (poster)" is an appropriate decision for this paper.

**Justification For Why Not Lower Score:**

All the reviewers are happy with accepting the paper, which I agree. In addition, the authors' response and the revised version of the paper have adequately addressed all major concerns. Therefore I recommend accepting the paper.

---

### Decision · Program_Chairs · 2024-01-16

Accept (poster)